# Identifying and interpreting tuning dimensions in deep networks

**Nolan S. Dey**
University of Waterloo
Vector Institute for AI
nsdey@uwaterloo.ca

**J. Eric Taylor**
Vector Institute for AI
University of Guelph
eric.taylor@vectorinstitute.ai

**Bryan P. Tripp**
University of Waterloo
bptripp@uwaterloo.ca

**Alexander Wong**
University of Waterloo
alexander.wong@uwaterloo.ca

**Graham W. Taylor**
University of Guelph
Vector Institute for AI
gwtaylor@uoguelph.ca

## Abstract

In neuroscience, a tuning dimension is a stimulus attribute that accounts for much of the activation variance of a group of neurons. These are commonly used to decipher the responses of such groups. While researchers have attempted to manually identify an analogue to these tuning dimensions in deep neural networks, we are unaware of an automatic way to discover them. This work contributes an unsupervised framework for identifying and interpreting "tuning dimensions" in deep networks. Our method correctly identifies the tuning dimensions of a synthetic Gabor filter bank and tuning dimensions of the first two layers of InceptionV1 trained on ImageNet.

## 1 Introduction

In neuroscience, tuning dimensions are foundational methods for understanding groups of biological neurons because they can provide a decipherable explanation of what groups of neurons respond to [7, 8, 14]. A tuning dimension is a stimulus attribute that accounts for much of the activation variance of a group of neurons. Since data collection in biological neurons is expensive and time consuming, the tuning dimensions for a group of neurons usually become clear over decades following painstaking trial-and-error.

In deep learning, these data collection limitations do not exist because we can easily measure neural activations for very large and diverse datasets. Some works have manually found an analogue to tuning dimensions in deep neural networks (DNNs) [2, 3]; however, there is currently no way to automatically identify them. In this paper, we define such a method as follows. First, we acquire a dataset of activations. Then, we apply unsupervised dimensionality reduction techniques to identify a set of basis vectors that explain that activation space's variance. Finally, we visualize points [15] along each basis vector to produce the tuning dimension. Our method identifies the tuning dimensions of a synthetic Gabor filter bank and our hypothesized tuning dimensions of the first two layers of InceptionV1 [23].

## 2 Related work

A popular approach to explaining the function of neurons in convolutional neural networks (CNNs) is to visualize their preferred input. These methods are compelling because the resultant images

2nd Workshop on Shared Visual Representations in Human and Machine Intelligence (SVRHM), NeurIPS 2020.

are highly interpretable and because the explanations can be generated automatically. A neuron's preferred stimulus can be visualized by optimizing an input image to maximize the activation of a particular neuron [11–13, 15, 16]. Some neurons show strong activation in response to a diverse range of stimuli [12, 15]. Furthermore, neurons rarely fire in isolation — multiple neurons often fire together to represent concepts [16]. Together, these findings suggest that studying the preferred stimulus of individual neurons alone is not sufficient, and sometimes misleading, for understanding deep representations.

To gain a more complete understanding of deep representations, we should strive to understand the activation space of each layer. Once trained, layers in deep neural networks transform the activation space of the previous layer in a way that minimizes some loss function. A point in activation space $\mathbf{a}$ can be visualized by optimizing an input image $I$ to approximately reproduce $\mathbf{a}$ when $I$ is forward propagated through the network [15]. It is difficult to study activation space because it is high-dimensional, even with a visualization method. [3, 10] project the high-dimensional activation space to a two-dimensional manifold while attempting to preserve high-dimensional distances. Then, points on the two-dimensional manifold can be sampled and visualized either through optimization [3] or by displaying the input data corresponding to each point [10]. These methods help us understand deep representations by visualizing the embedding distances between visual concepts. However, by compressing high dimensional activation spaces to two dimensions, a great deal of information is lost. When identifying tuning dimensions, we need not constrain ourselves to compressing activation space to two or three dimensions because a layer can have more than three tuning dimensions.

There has also been some work on understanding deep representations by manually identifying tuning dimensions in deep networks. For layer mixed4c in InceptionV1 [23], Carter et al. [3] manually found three paths along the two-dimensional manifold corresponding to a change in a stimulus attribute e.g. number of fruit, number of people, and blur level of foliage. These paths are tuning dimensions because they correspond to a change in a stimulus attribute. This result shows that it is possible to find semantically meaningful tuning dimensions based on a dataset of activations. Cammarata et al. [2] manually found that the angular orientation of curves in the input is a tuning dimension for curve detector neurons in multiple InceptionV1 [23] layers. In other words, the activation of multiple neurons can be explained by a single stimulus attribute. This result demonstrates the value of tuning dimensions for producing decipherable explanations of the behaviour of groups of neurons. Though this manual process of identifying tuning dimensions bears some similarity to neuroscience, it is a time consuming process. We are unaware of any existing methods for automatically identifying tuning dimensions in DNNs.

## 3 Method

In this paper we investigate layers of InceptionV1 [23] pretrained on ImageNet [5]. Prior works have also studied this combination [2, 3, 15–17]. ImageNet is a large, diverse sample of natural images and InceptionV1 was previously state-of-the-art for the ImageNet classification task [22].

### 3.1 Collecting activations

We first obtain a representative sample of a layer's activations. This is the same method of collecting activations used by Carter et al. [3]. We forward propagate $N$ randomly chosen ImageNet [5] images through InceptionV1 up to a specified layer to obtain an $(N \times h \times w \times c)$ matrix of activations, where $h$ and $w$ are the spatial height and width, and $c$ is the number of channels. Following [3], we found $N = 1\,\mathrm{M}$ to be sufficient although it is likely that a much lower $N$ would also be sufficient. If we collected activations from the center spatial position, we would be biasing our collection towards input features typically found at the center of images. To obtain a representative sample of activation space while also respecting memory constraints, we choose a random spatial index (padded by one spatial position to avoid boundary artifacts) to collect an activation vector ($\mathbb{R}^c$) for each image to obtain activations $A \in \mathbb{R}^{N \times c}$.

### 3.2 Identifying tuning dimensions

We apply a dimensionality reduction technique to $\mathbf{a}$ to obtain a set of lower dimensional transformed activations $A' \in \mathbb{R}^{N \times n}$ where $n$ is the number of reduced dimensions. We treat each of the $n$

dimensions of $A'$ as tuning dimensions. We compare the scikit-learn [19] implementations of several dimensionality reduction techniques in Section 4.1: principal component analysis (PCA), independent component analysis (ICA) [9], non-negative matrix factorization (NMF) [4], and locally linear embedding (LLE) [21].

### 3.3 Visualizing points along each tuning dimension

To visualize a tuning dimension, we sample $m$ points ($\mathbb{R}^n$) along the tuning dimension to obtain an $(m \times n)$ matrix of sampled points in transformed activation space. Then, we apply an inverse transform to obtain an $(m \times c)$ matrix of sampled points in untransformed activation space. We found that uniformly sampling $m$ points between the observed minimum and maximum values along each tuning dimension produced the most informative visualizations. Sampling methods are compared in Section A.1. j Let $\mathbf{a}' \in \mathbb{R}^{h \times w \times c}$ be the activation obtained from forward propagating an image parameterization $I$ up to a specified layer. Following Olah et al. [15], we visualize a sampled point in activation space $\mathbf{a} \in \mathbb{R}^c$ by optimizing an image parameterization $I$ to maximize the mean cosine similarity between $\mathbf{a}$ and $\mathbf{a}'$ as follows:

$$I^* = \arg\max_I \Big( \frac{1}{wh} \sum_i^w \sum_j^h \frac{\mathbf{a} \cdot \mathbf{a}'_{ij}}{\|\mathbf{a}\| \|\mathbf{a}'_{ij}\|} \Big) \tag{1}$$

This optimization results in an image $I^*$ which approximately reproduces $\mathbf{a}$ at every spatial location in the activation map $\mathbf{a}'$ of the intended layer. We also tried optimizing an image $I$ to reproduce $\mathbf{a}$ at a single spatial location in $\mathbf{a}'$ but this resulted in less interpretable visualizations.

The torch-lucent library was used for visualization.[1] Following Olah et al. [15], we applied small random affine transformations to $I$ each optimization step, and the image parameterization $I$ we used was the Fourier basis with all frequencies scaled to have the same energy. When we used a pixel-based image parameterization $I$ instead, the resulting visualizations $I*$ resembled noise. However, prior work suggests that if we were visualizing the activations of a network that was trained with an adversarial robustness objective, optimizing a pixel-based image parameterization $I$ should yield more interpretable visualizations [6].

### 3.4 Interpreting tuning dimension visualizations

A tuning dimension is a stimulus attribute that accounts for much of the activation variance of a group of neurons. While studying a tuning dimension visualization, a change in a stimulus attribute along the dimension is expected. The interpretation of the changing stimulus attribute(s) is a subjective process. We are more likely to notice changes in a hypothesized stimulus attribute, which introduces the danger of confirmation bias. To provide transparency, our interpretations are included in the Appendix sections A.3 and A.4.

Points in $A'$ obtained from PCA or ICA can take both positive and negative values. When examining points along a dimension of $A'$ obtained from PCA or ICA, the points at the positive and negative ends represent opposite concepts, while the points near zero represent the mean collected activation vector. Figure 1 shows visualizations of two tuning dimensions identified by ICA for `conv2d0`, the first layer of InceptionV1 [23]. Points in $A'$ obtained from NMF can only take positive values.

## 4 Results

In Section 4.1, we compare dimensionality reduction models and demonstrate that our method can correctly identify the tuning dimensions of a synthetic Gabor bank. In Section 4.2, we study the tuning dimensions of the first two layers of InceptionV1 [23] and show that our method can identify a set of hypothesized tuning dimensions. We found it best to study these visualizations through an interactive interface.[2]

---

[1]https://github.com/greentfrapp/lucent
[2]Link to interactive demo: `https://tinyurl.com/tuning-dimensions-svrhm`

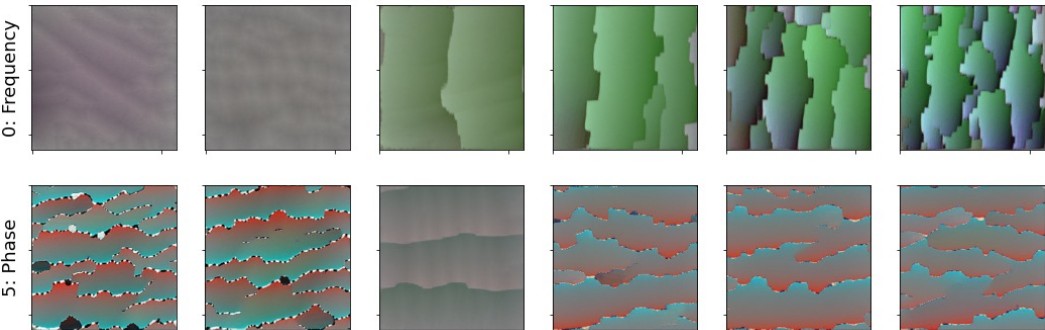

Figure 1: Tuning dimension visualizations of the first layer of InceptionV1 (conv2d0). Top: Visualized points along first ICA component show increasing frequency of vertically oriented edges. Bottom: Visualized points along sixth ICA component show dark/light edges switching to light/dark edges which is associated with a change in Gabor filter phase. 32 points were visualized for each tuning dimension but to improve readability, the most informative 6 are shown. Best viewed in color.

## 4.1 Identifying tuning dimensions of a synthetic Gabor bank

To compare dimensionality reduction methods for identifying tuning dimensions, we need a method to validate the recovery of known tuning dimensions. To ensure the correct tuning dimensions are known *a priori*, we construct a bank of Gabor filters with tuning dimensions of frequency, angle, and phase. Both the human visual system and trained CNNs contain Gabor-like filters in early layers [17, 18]. Frequency, angle, and phase affect the filter's preferred edge frequency, edge angle, and dark/light edge orientation respectively. Since conv2d0, the first layer of InceptionV1 [23], has 28 $7 \times 7$ Gabor filters [17] applied with a stride of 2, we used the same number of filters, kernel size, and stride for the Gabor filter bank. More details regarding the Gabor filter bank's construction are included in the Appendix section A.2.

Table 1: Comparison of identified Gabor bank tuning dimensions. PCA-3 denotes PCA with 3 components.

| Tuning Dimension | PCA-3 | PCA-6 | ICA-3 | ICA-6 | NMF-3 | NMF-6 | LLE-3 | LLE-6 |
|---|---|---|---|---|---|---|---|---|
| Frequency | ✓ | ✓ | ✓ | ✓ | ✗ | ✗ | ✗ | ✗ |
| Angle | ✗ | ✗ | ✓ | ✓ | ✗ | ✗ | ✗ | ✗ |
| Phase | ✓ | ✓ | ✓ | ✓ | ✗ | ✗ | ✗ | ✗ |

We collected activations and applied standard unsupervised learning techniques to identify the tuning dimensions: PCA, ICA [9], NMF [4], and LLE [21]. Table 1 shows that only ICA correctly identified all of the tuning dimensions, PCA failed to identify angle, and both NMF and LLE failed to identify any of the tuning dimensions. PCA may have failed to identify the angle tuning dimension because each component is constrained to be orthogonal whereas ICA may have succeeded because it does not have this constraint. The visualizations of points along each NMF tuning dimension did not have any meaningful variation. This could be due to the NMF constraint that points along each tuning dimension be non-negative. Visualized points along each LLE tuning dimension were very different but had no apparent pattern between them. We continue to use both PCA and ICA because they identified most of the Gabor filter bank tuning dimensions. We wanted a method that is not sensitive to hyperparameters so we did not try deep learning-based dimensionality reduction techniques.

## 4.2 Identifying tuning dimensions of InceptionV1 layers

There is no quantitative metric to evaluate the validity of tuning dimensions. Fortunately, Olah et al. [17] published a study of individual neurons in the early layers of InceptionV1 which could be used as a reference to hypothesize the ground-truth tuning dimensions in this particular architecture. When evaluating different variations of our method, we compared our results to the tuning dimensions we had hypothesized. In other words, we ask whether our method for automatic discovery of tuning dimensions converges on the same dimensions manually identified by other researchers.

Table 2: Comparison of PCA and ICA for identifying tuning dimensions in the InceptionV1 layers `conv2d0` and `conv2d1`. PCA-16 denotes PCA with 16 components.

| | conv2d0 | | conv2d1 | |
| Hypothesized Tuning Dimension | PCA-16 | ICA-16 | PCA-16 | ICA-16 |
| --- | --- | --- | --- | --- |
| Gabor frequency | ✗ | ✓ | ✓ | ✓ |
| Gabor angle | ✓ | ✓ | ✓ | ✓ |
| Gabor phase | ✓ | ✓ | ✓ | ✓ |
| Green/purple contrast | ✓ | ✓ | ✓ | ✓ |
| Orange/blue contrast | ✓ | ✓ | ✓ | ✓ |
| Green/red contrast | ✗ | ✓ | ✗ | ✓ |
| Blue/red contrast | ✓ | ✓ | ✓ | ✓ |

**conv2d0**

Olah et al. [17] manually categorized the `conv2d0` neurons into Gabor, color contrast, and uncategorized families. We hypothesized that the Gabor neurons would have the same tuning dimensions as our synthetic Gabor bank: frequency, angle, and phase (dark/light orientation of edges). Based on the color contrast neurons, we expected to see green/purple, orange/blue, green/red, and blue/red color contrasts. We did not hypothesize any tuning dimensions for the uncategorized units. After fitting ICA with 16 components and visualizing the tuning dimensions, all of the hypothesized tuning dimensions were identified (Table 2). Each of the visualizations that we interpreted as one of the hypothesized tuning dimensions is included in Section A.4.

**conv2d1**

Olah et al. [17] manually categorized the `conv2d1` neurons into low frequency, Gabor, color contrast, complex Gabor, multicolor, color, hatch, and uncategorized families. For the Gabor and color contrast neurons we hypothesized the same tuning dimensions as `conv2d0`. We also expected the complex Gabor filters to have frequency and angular orientation tuning. We did not hypothesize any tuning dimensions for the remaining neuron categories. ICA with 16 components identified all of the hypothesized tuning dimensions (Table 2). Each of the visualizations that we interpreted as one of the hypothesized tuning dimensions is included in Section A.4.

## 5 Conclusions and future work

We propose the first general unsupervised method for identifying and visualizing the tuning dimensions of a deep neural network layer. We show the method correctly identifies the tuning dimensions of a synthetic Gabor filter bank and hypothesized tuning dimensions of the first two layers of InceptionV1. Moving forward, we have identified a number of areas for future work.

A key area of future work will studying the higher layers of InceptionV1 in more detail. It takes approximately 2 minutes to manually interpret each tuning dimension. The layers after `conv2d1` have many neurons (>= 192), making it more time consuming to study these layers. The higher layers of InceptionV1 [23] also contain more diverse and complex neurons, making the tuning dimensions of each layer unclear. Without clear hypotheses for a layer's tuning dimensions, this introduces a great deal of uncertainty into the qualitative evaluation of the tuning dimensions.

Our method is not specific to a particular data instance, making it a global explainability method. Once our method is used to label the tuning dimensions of a layer, the labels may be used in a "what-if" tool [24] to provide instance-specific explanations. We may also use an external dataset such as Broden [1] to quantitatively associate each tuning dimension with specific visual concepts.

Finally, our method may be extended to identify tuning dimensions in reinforcement learning, natural language processing, and graph learning models provided a method exists to visualize an activation **a** by optimizing an input parameterization to reproduce **a**. Ponce et al. [20] proposed a method for optimizing the preferred stimuli $I*$ of individual biological neurons. Their method could be extended to optimize stimuli $I*$ which reproduce a particular activation vector **a**. This extension could allow our method to be applied to biological neural networks.

## Acknowledgments and Disclosure of Funding

Thank you to BMO Bank of Montreal for funding this research.

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

# A Appendix

## A.1 Comparing sampling methods

When exploring visualization methods there was no metric for interpretability so we qualitatively evaluated the visualizations based on their perceived interpretability. We found that sampling and visualizing 32 points along each tuning dimension was sufficient for observing the differences in the visualizations along the dimension.

We found that uniformly sampling $m$ points between the observed minimum and maximum values along each tuning dimension produced the most informative visualizations. We also tried sampling so that the same proportion of observed points in activation space lie between each of the points along the tuning dimension. Since most activations are zero or near zero, this sampling strategy resulted in over-sampling the region near zero. Visualizations produced using these sampling methods are shown in Figure 2.

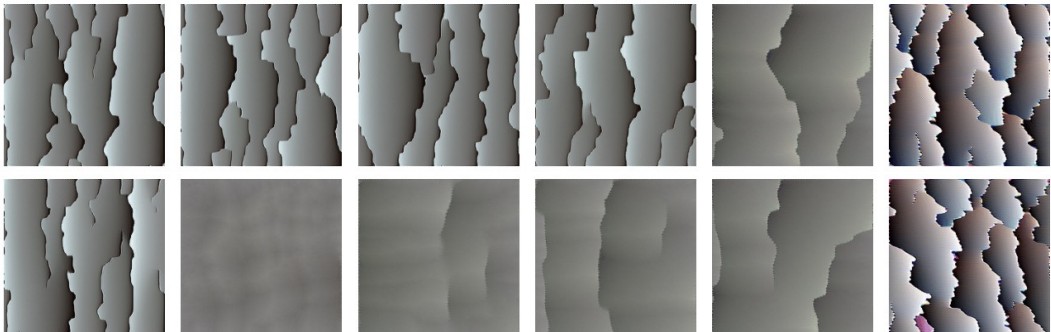

Figure 2: Visualizations of the first ICA component fit to activations from the first layer of InceptionV1 (`conv2d0`). Top: Points sampled uniformly between the observed minimum and maximum values along the component. Bottom: Points sampled such that the same proportion of observed points in activation space lie between each of the points along the component. 32 points were visualized for each tuning dimension but only 6 are shown to improve readability. Best viewed in color.

## A.2 Gabor bank construction

A Gabor filter $g$ is constructed by multiplying a sinusoid with a Gaussian as shown in Equation 2, where $x' = x \cos \theta + y \sin \theta$ and $y' = -x \sin \theta + y \cos \theta$.

$$g(x, y; \lambda, \theta, \psi, \sigma, \gamma) = \exp(-\frac{x'^2 + \gamma^2 y'^2}{2\sigma^2}) \cos(2\pi \frac{x'}{\lambda} + \psi) \tag{2}$$

The Gabor bank was constructed by uniformly sampling (with a fixed random seed) frequency $(1/\lambda) \in [0.2, 0.3]$, angle $\theta \in [0, \frac{\pi}{2}]$, and phase $\psi \in [\frac{-\pi}{2}, \frac{\pi}{2}]$. We set $\gamma = 1$ and $\sigma = 2$. The Gabor filter weights and preferred stimuli are shown in Figure 3.

## A.3 Gabor bank tuning dimension visualizations

As discussed in Section 3.4, the visualizations must be interpreted by a human. The interpretation can be subjective so for transparency, we have included each of the visualizations that we interpreted as tuning dimensions. The results can also be explored through our interactive interface. The tuning dimension visualizations for PCA-3 and PCA-6 (Figure 4), ICA-3 (Figure 5), ICA-6 (Figure 6), NMF-3 (Figure 7), and LLE-3 (Figure 8) are included.

## A.4 InceptionV1 tuning dimension visualizations

As discussed in Section 3.4, the visualizations must be interpreted by a human. The interpretation can be subjective so for transparency, we have included each of the visualizations that we interpreted as tuning dimensions. The results can also be explored through our interactive interface. The tuning dimension visualizations for `conv2d0` PCA-16 (Figure 9), `conv2d0` ICA-16 (Figure 10), `conv2d1` PCA-16 (Figure 11), and `conv2d1` ICA-16 (Figure 12) are included.

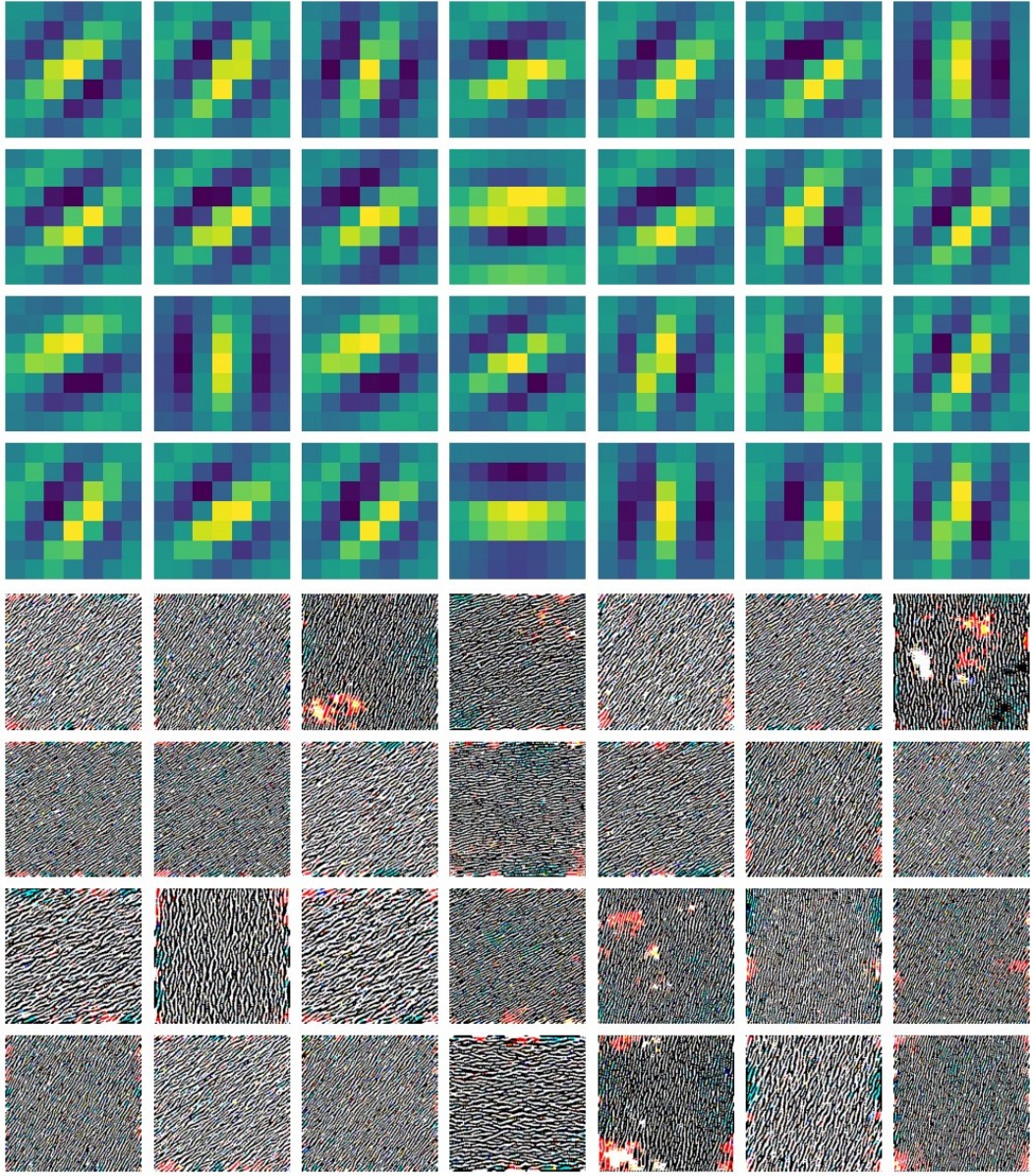

Figure 3: Top: Gabor bank filter weights. The values in each of the RGB channels are equal. Bottom: Preferred stimuli of each Gabor filter found through feature visualization [15]. Best viewed in color.

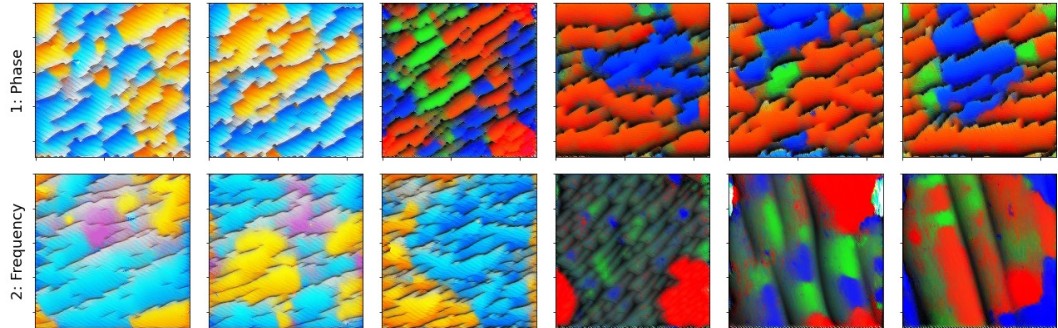

Figure 4: Gabor bank PCA-3 and PCA-6 tuning dimension visualizations with their component number and interpreted tuning dimension shown on the left. 32 points were visualized for each tuning dimension but only 6 are shown to improve readability. Best viewed in color.

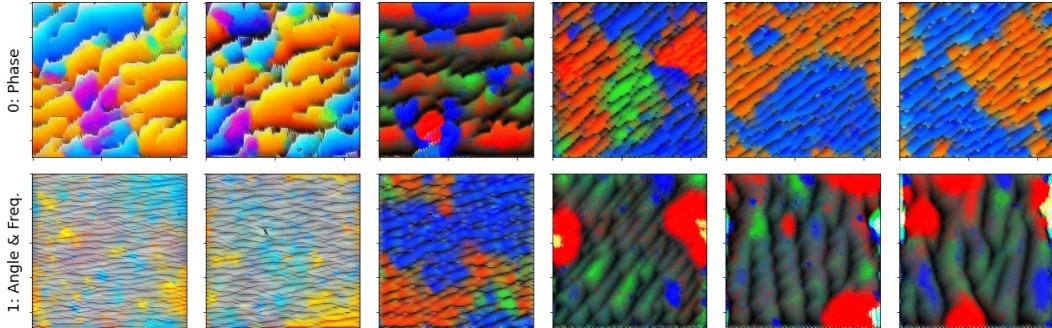

Figure 5: Gabor bank ICA-3 tuning dimension visualizations with their component number and interpreted tuning dimension shown on the left. 32 points were visualized for each tuning dimension but only 6 are shown to improve readability. Best viewed in color.

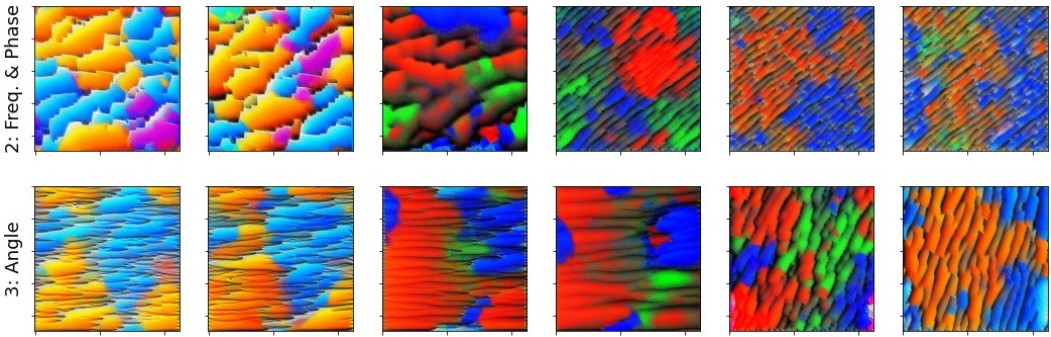

Figure 6: Gabor bank ICA-6 tuning dimension visualizations with their component number and interpreted tuning dimension shown on the left. 32 points were visualized for each tuning dimension but only 6 are shown to improve readability. Best viewed in color.

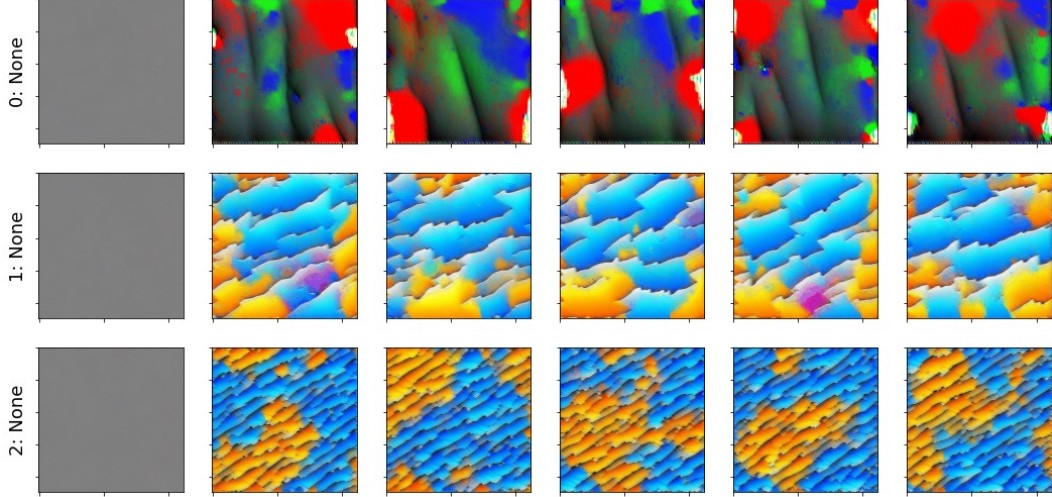

Figure 7: Gabor bank NMF-3 tuning dimension visualizations with their component number and interpreted tuning dimension shown on the left. No meaningful variation was observed along each tuning dimension so no tuning dimensions were interpreted. 32 points were visualized for each tuning dimension but only 6 are shown to improve readability. Best viewed in color.

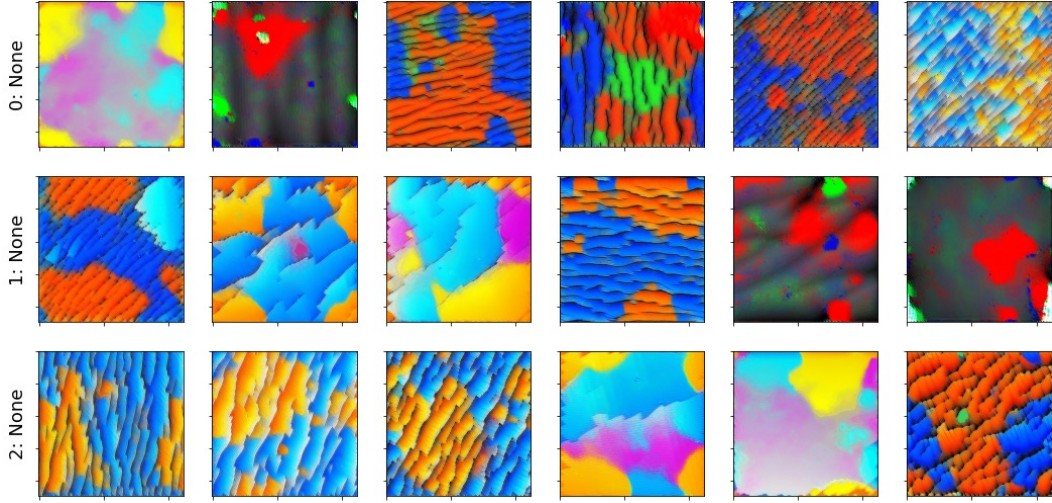

Figure 8: Gabor bank LLE-3 tuning dimension visualizations with their component number and interpreted tuning dimension shown on the left. No apparent pattern was observed along each tuning dimension so no tuning dimensions were interpreted. 32 points were visualized for each tuning dimension but only 6 are shown to improve readability. Best viewed in color.

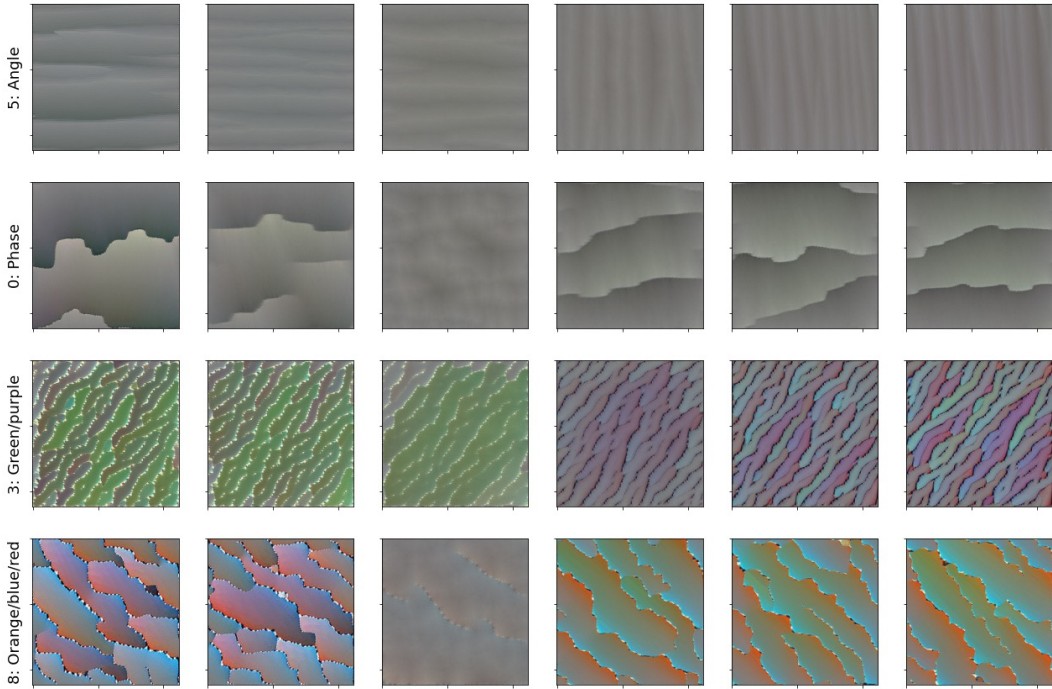

Figure 9: `conv2d0` PCA-16 tuning dimension visualizations with their component number and interpreted tuning dimension shown on the left. 32 points were visualized for each tuning dimension but only 6 are shown to improve readability. Best viewed in color.

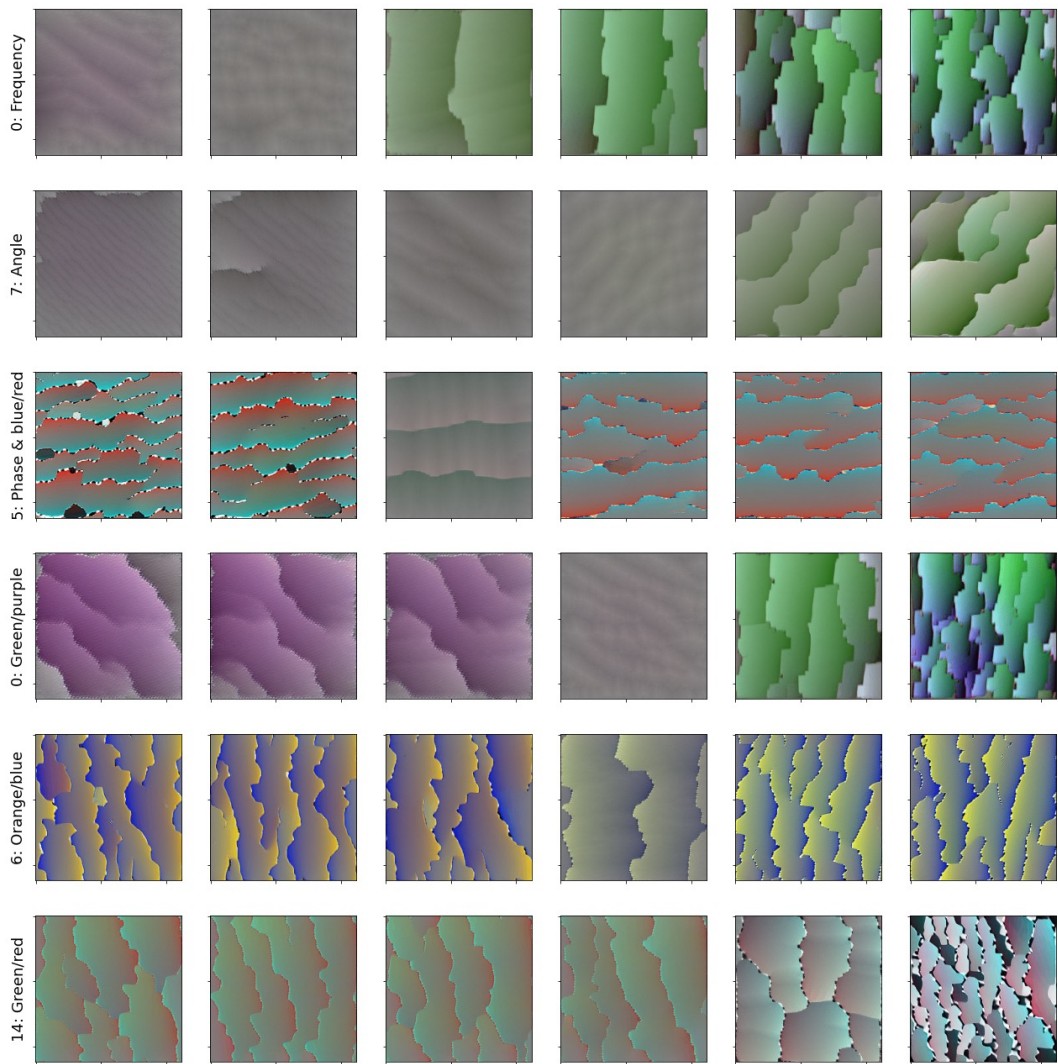

Figure 10: `conv2d0` ICA-16 tuning dimension visualizations with their component number and interpreted tuning dimension shown on the left. 32 points were visualized for each tuning dimension but only 6 are shown to improve readability. Best viewed in color.

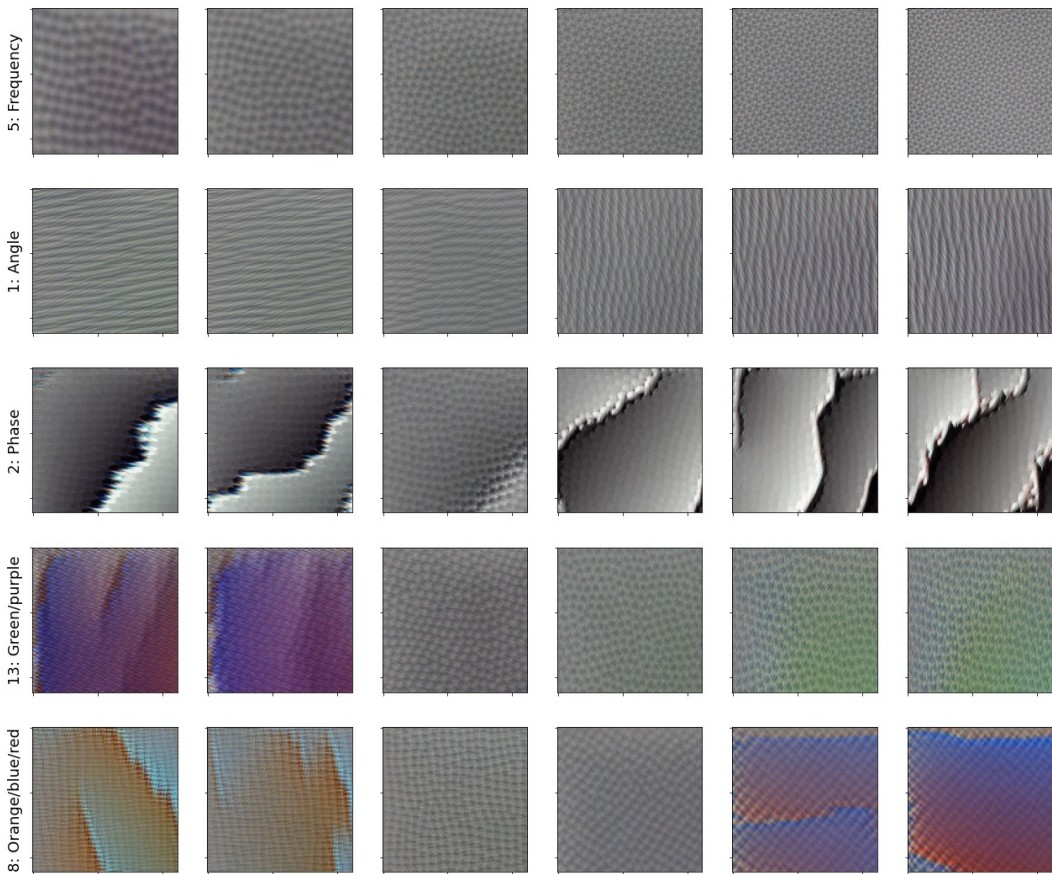

Figure 11: `conv2d1` PCA-16 tuning dimension visualizations with their component number and interpreted tuning dimension shown on the left. 32 points were visualized for each tuning dimension but only 6 are shown to improve readability. Best viewed in color.

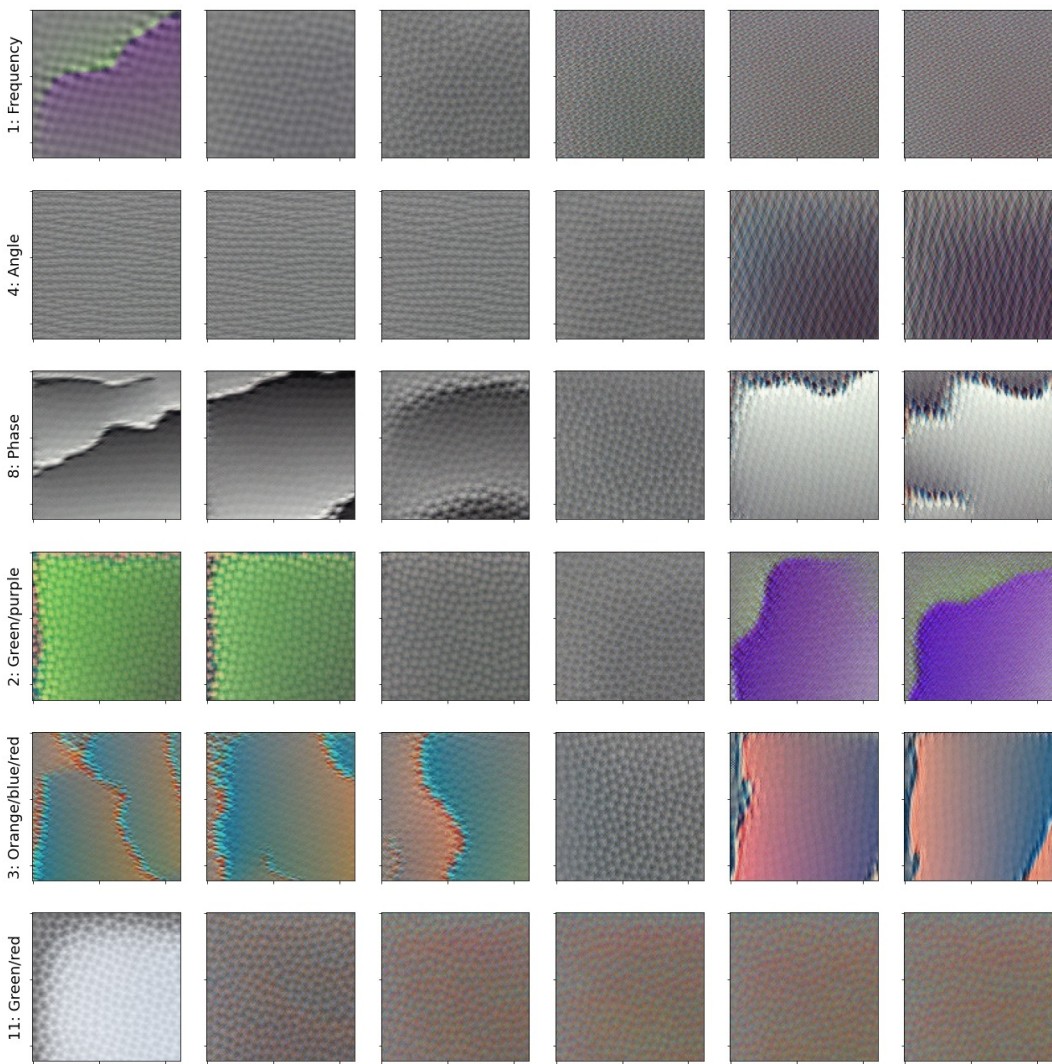

Figure 12: `conv2d1` ICA-16 tuning dimension visualizations with their component number and interpreted tuning dimension shown on the left. 32 points were visualized for each tuning dimension but only 6 are shown to improve readability. Best viewed in color.

