# OpenReview forum: "Identifying and interpreting tuning dimensions in deep networks"
_NeurIPS.cc/2020/Workshop/SVRHM — SVRHM@NeurIPS Poster_

### Official Review · AnonReviewer2 · 2020-10-27
**Identifying and interpreting tuning dimensions in deep networks**

**Rating:** 7
**Confidence:** 2

**Review:**

The authors have developed an unsupervised method to visualize tuning dimensions of individual units in a deep neural network. While previous work exploring tuning profile relied on manually identifying them, this is the first automatized approach to exploring tuning dimensions. They test their approach on a synthetic Gabor filter bank and on the first layers of InceptionV1.

Strengths:
- Strengths: As far as I am aware, this is a novel, automatized approach to exploring low-level tuning properties of early layers DNNs. The authors provide a good test of the model by employing a synthetic Gabor bank, and by comparing the dimensions identified by the model for early layers of InceptionV1 to dimensions that were identified by previous literature.

Weaknesses:
- at page 4, in section 3.4, the authors raise the issue of confirmation bias when the presence of an attribute that needs to be explored qualitatively is hypothesized, but aside from mentioning this, no solution or way to correct that is proposed. I wonder why they did not have multiple raters identify the property of a given component, and then decide on the emerging property based on agreement across raters.

Clarifications:
- in section 3.1, "from" should be removed in the sentence "from for each image to obtain activations..."
- To give a transparent account of the subjective interpretation of tuning dimensions, the authors provide visualization for all their interpretations in the Appendix. However, they do not reference the Appendix in the main text, where the limitations of the ratings are noted (page 3).
- In Figure 1, where along the dimensions are visualizations sampled from? Do they cover the whole range of the dimension tuning? While this information is present in the main text and Appendix, it would be helpful to provide it in the figure legend
- At page 7, Figure 2 is referenced, but it is not present in the Manuscript.

Clarity:
- Overall, the paper is understandable. However, some parts might benefit from some editing to improve clarity (see above comments).

---

### Official Review · AnonReviewer1 · 2020-10-28
**intriguing analysis of tuning dimensions, but seems preliminary**

**Rating:** 4
**Confidence:** 5

**Review:**

This paper presents an intriguing preliminary analysis trying to probe the “tuning dimensions” of neural networks via an unsupervised dimensionality reduction technique. Most previous work identifying tuning dimensions focused on manually looking at the maximal responses, or probing individual neurons, while the authors in this paper try to describe the dimensions of a full layer by applying dimensionality reduction techniques to a large set of activations from the layer. The authors compare a few different dimensionality reduction techniques including PCA, ICA, NMF, and LLE, and they validate the method by using a gabor filter bank and a previously published study which labels the tuning dimensions of an inception network by hand.

A poster on this topic at SVRHM may lead to interesting conversations about the representations of deep networks, however my highlighted technical concerns about the optimization process and the fit of the topic for the workshop leads me to recommend this paper for rejection.

Pros:
* I appreciated the discussion on whether individual units of a network should be studied, or whether ensembles of neurons should be considered instead.
* The authors demonstrate the success of the method by applying it to a Gabor filterbank, where the tuning dimensions are well known, and show that the method “finds” the expected dimensions
* The analysis of the representations is fairly simple to explain and understand -- simply collect the activations and perform standard dimensionality reduction techniques

Cons:
* Optimization details for the inverse transform are not well described, which might make future experiments in novel contexts difficult. This also raises questions about the generality of the presented results. Specifically, I questioned whether there was a prior applied during the optimization to make them “look” more realistic (for instance a smoothness prior, which I suspect is applied given the splotches of color in the example images) which can bias interpretation methods towards things that are “human-like” without these properties being present in the model (see discussions of this phenomena in “Adversarial Robustness as a Prior for Learned Representations” by Engstrom et al. 2019 and “Metamers of neural networks reveal divergence from human perceptual systems” by Feather et al. 2019). If such a prior is applied, how sensitive are the results towards this type of decision?
* A second technical concern about the optimization is how well it succeeded. There is no guarantee that a point in activation space will have a realizable input representation, and this problem may be worse with more non-linearities. A future submission would be strengthened by quantifying the optimization success.
* Why are only the first two layers of the inception network shown? The paper would be strengthened if all layers were analyzed.
* Contrary to the author’s description that the dimensions are intuitive to interpret (line 165 saying “it takes approximately 2 minutes to manually interpret each tuning dimension”), I found the visualizations difficult to understand without first reading the authors interpretation. Including an experiment on human validation of the labels would strengthen the paper.
* The main contribution of the paper seems to be the analysis of neural networks, and there is little comparison with biological representations (other than the baseline gabor filters being based on biology). Given that SVRHM is about representations of humans and machines, I am unsure that a pure analysis on the interpretation of neural networks is a good fit.

---

### Official Review · AnonReviewer3 · 2020-10-28
**Unsupervised estimation of tuning dimensions for both Gabor Filter Bank and InceptionV1**

**Rating:** 8
**Confidence:** 5

**Review:**

Summary:

This work expanded on Olah et al, by introducing an unbiased estimation of tuning dimensions for deep neural networks. This is a promising direction for the interpretation of neural networks.

Pros:

- Simple estimation of tuning dimensions via dimensionality reduction.
- Proof of concept with Gabor Filter Bank, which showed PCA and ICA to be the best dimensionality reduction techniques for this analysis.
- Validation of findings from Olah et al with InceptionV1 model.


Recommendations:
- Test robustness of the method to different instances of the same model (different initializations of InceptionV1)
- Expand this analysis to other models.
- Expand this work to higher layers of the model (as suggested by the paper)
- Resample activation space and compare tuning dimensions of this new subspace.

---

### Public Comment · ~Nolan_Simran_Dey1 · 2020-12-08
**Response to reviews**

We thank all of the reviewers for their helpful and thoughtful comments. To address some of the reviewer comments we:

- Included more details about the feature visualization optimization in section 3.3
- Described how our method could be extended to study biological neural networks
- Fixed typos and any other minor issues brought up by the reviewers
- Referenced A.3 and A.4 in section 3.4

There were many excellent suggestions for improvement that are beyond the scope of this workshop submission. Those we have left for future work.

---

### Decision · Program_Chairs · 2020-11-02

Accept (Poster)